# Towards Development of Molecularly Imprinted Electrochemical Sensors for Food and Drug Safety: Progress and Trends

**DOI:** 10.3390/bios12060369

**Published:** 2022-05-27

**Authors:** Shuhong Zhou, Chen Liu, Jianguo Lin, Zhi Zhu, Bing Hu, Long Wu

**Affiliations:** 1Key Laboratory of Fermentation Engineering (Ministry of Education), College of Bioengineering and Food, Hubei University of Technology, Wuhan 430068, China; zhouwx777@163.com (S.Z.); jianguolin@hbut.edu.cn (J.L.); 2Leibniz-Institute of Photonic Technology, Leibniz Research Alliance-Leibniz Health Technologies, Albert-Einstein-Str. 9, 07745 Jena, Germany; chen.liu@leibniz-ipht.de; 3Key Laboratory of Tropical and Vegetables Quality and Safety for State Market Regulation, School of Food Science and Engineering, Hainan University, Haikou 570228, China; z0108150427@163.com; 4Key Laboratory of Biotechnology and Bioresources Utilization of Ministry of Education, School of Life Sciences, Dalian Minzu University, Dalian 116600, China; hubing19871121@163.com

**Keywords:** electrochemical sensors, molecularly imprinted polymers, separation and detection, food safety, antibiotics detection

## Abstract

Due to their advantages of good flexibility, low cost, simple operations, and small equipment size, electrochemical sensors have been commonly employed in food safety. However, when they are applied to detect various food or drug samples, their stability and specificity can be greatly influenced by the complex matrix. By combining electrochemical sensors with molecular imprinting techniques (MIT), they will be endowed with new functions of specific recognition and separation, which make them powerful tools in analytical fields. MIT-based electrochemical sensors (MIECs) require preparing or modifying molecularly imprinted polymers (MIPs) on the electrode surface. In this review, we explored different MIECs regarding the design, working principle and functions. Additionally, the applications of MIECs in food and drug safety were discussed, as well as the challenges and prospects for developing new electrochemical methods. The strengths and weaknesses of MIECs including low stability and electrode fouling are discussed to indicate the research direction for future electrochemical sensors.

## 1. Introduction

Since food and drug safety are closely related to human health, it is essential to develop rapid and effective analytical methods to monitor food and drugs and control their safety. Electrochemical sensors are a class of chemical sensors in which an electrode is used as a transducer to give an electrochemical signal for analytes [1,2]. Most electrochemical sensors are small-sized devices with simple operations and rapid detection, which make them suitable for on-site applications and popular in the analysis of food and drugs [3,4]. However, electrochemical sensors suffer from some limitations when they are applied in food and drug detection: (1) it is difficult to achieve high sensitivity and accuracy due to the lack of effective standards for constructing the sensors; (2) the complex sample matrix has great effects on the stability and reproducibility of the electrochemical signal output. Therefore, there is a great need to improve electrochemical sensors to meet the requirements in food and drug safety detection.

To achieve higher performance of electrochemical sensors, numerous studies have been carried out, mainly including the development of new electrodes, modification of electrodes, construction of detection modes, and design of signal labels [5,6,7,8]. To overcome the limitations associated with high costs and complex processing procedures of traditional electrodes (glassy carbon electrodes, silver, and gold electrodes, etc.), simple screen-printed electrodes [9], indium tin oxide (ITO) electrodes [10], paper-based electrodes [11] and micro-electrodes [12], have been developed in different applications. Usually, they cannot achieve desirable sensitivity due to their limited specific surface area. So, nanomaterials with various functions are widely used to modify the electrodes and improve detection sensitivity [13]. However, most of the modifications are performed by simply adsorbing the nanomaterials on the electrode surface to obtain a large specific surface area and efficient electron transfer rate [14,15], and the physical adsorption is not reliable enough to give stable signals, thus it will result in a low reproducibility in measurements. For the detection mode, different electrochemical sensors can be constructed according to the distinct receptors of enzymes, antibodies, and aptamers, including enzyme sensors, immunoassays, and DNA sensors, to meet the requirements of analytes in complex samples [16,17,18,19]. Commonly, the sensing structures are built by physical adsorption depending on the interaction of Van der Waals force between receptors and electrode surface, which readily gives rise to non-specific adsorption with weak adsorption, and thus could lead to a decrease in the overall stability of electrochemical sensors [20,21]. The last, but not least, electrochemical signal label is closely related to detection sensitivity and accuracy, and the design of the signal includes an electroactive substance and signal amplification strategy. To indicate analytes, various electroactive substances are used as electrochemical signals with signal amplification to improve detection sensitivity [22,23]. However, the single-signal output is prone to be influenced by external environmental interferences, so it is necessary to design the signals to eliminate the interferences and enhance the sensitivity. 

The above-mentioned four aspects describe the electrochemical sensors from the development of the electrode to the design of the signal. Unfortunately, current work mainly focuses on improving one part of them but ignores the importance of studying the four aspects as a whole [24,25,26]. Since the electrode interface, recognition element and signal outputs are the unified whole of electrochemical sensors, it is difficult to achieve desired detection performance by only enhancing one part, either developing new electrodes or constructing a detection mode [27,28,29]. That is, only by considering mutual influences from different aspects, can it be possible to construct stable and reliable electrochemical sensing methods. The molecular imprinting technique (MIT) coupled with electrochemical sensors provides an opportunity to address the above concerns. As it is known that MIPs are characterized by predetermined structure-activity, specific recognition, and wide practicability [30,31]. As MIPs have specific recognition for target molecules, they can act as receptors to construct MIT-based electrochemical sensors (MIECs) with high specificity [32]. However, the relatively low adsorption capacity and mechanic properties of MIPs have become another challenge. It was reported that nanomaterials can effectively increase the specific surface area of MIPs and improve their conductivity, which makes MIECs more sensitive and stable [33,34]. Besides that, by adding nanomaterials such as gold nanoparticles (AuNP) into MIPs, they can provide an internal reference signal, and thus reduce the background interference [35]. In this regard, nanomaterials play a vital role in constructing MIECs with stable and accurate electrochemical signals. 

Based on the above discussions, though MIECs suffer from the limitations such as low conductivity and stability, they are still one of the few sensors that can hopefully improve the detection performance in whole aspects. With careful design and modifications, the setbacks of MIECs can be overcome to a large extent. In general, MIECs can be regarded as new kinds of analytical methods that readily realize sensitive, specific, accurate and rapid detection, with the integrated improvement from electrodes to signals. This review summarizes all kinds of MIECs as well as the preparation of MIPs, design of electrochemical signals, functions of MIECs and applications in food and drug safety, aiming to present a general comment on the development of MIECs and make a bridge between MIPs and electrochemical sensors (Figure 1). Furthermore, critical perspectives and discussions are given on the current progress and trends of MIECs and their application prospects. 

## 2. Brief Introduction of MIECs

### 2.1. Principle of MIECs

MIECs are integrated sensing techniques that use MIPs as recognition elements and electrochemical platforms as signal transducers (Figure 2). When the MIPs are well-prepared on the surface of the electrode, they can specifically recognize the analyte and indicate the molecular information (structure, concentration, conductivity, etc.) based on different electrochemical signals (CV for cyclic voltammetry, EIS for electrochemical impedance spectroscopy, SWV for Square-wave voltammetry, DPV for differential pulse voltammetry). As the analytes bind with MIPs, the direct detection can be achieved by the change in the state of MIPs or surface potential, which is due to a change in properties of the system without any electrodic reaction of analytes. As the specificity of MIPs towards analytes, the changes are only dependent on the amounts of analytes, and thus can be measured by the CV or EIS with variations of current or resistance. For example, Hedborg et al. reported the first MIPs-based capacitance sensor with capacitive or impedance detection, which is based on the principle of plate-capacitor with double layer phenomenon [36]. When analytes rebind with MIPs, the capacitance will vary with the concentration of analytes. This impedometric binding detection can be achieved by MIECs without the electrodic reaction of analytes.

Furthermore, if the analytes are electroactive substances, they will undergo an electrodic reaction when applied with an electrochemical technique such as SWV or DPV. By this means, the analytes will show a redox peak with a specific location. Moreover, the redox peak current will vary with the concentrations of analytes. In the label-free measurements, MIPs only act as receptors for the specific recognition with analytes. In a word, the MIECs combine the working principle of MIPs and that of electrochemical sensors to achieve detection specificity and sensitivity.

### 2.2. The Classification of MIECs

Since the electrochemical signals are output through the electrodes connected system, the electrodes used in MIECs can be varied with different applications. With the modification of MIPs on the electrode, either via an ex-situ or in-situ method, different kinds of electrodes vary in their functions. Besides the commonly used glass carbon electrode (GCE) and gold electrode, many other electrodes such as the ITO electrode, screen-printed electrode (SPE), and paper-based electrode are developed owing to the low cost and easy preparation. Thus, MIPs can be modified on different electrodes to achieve different operations. For example, on an ITO electrode, a researcher can conduct the surface pretreatment to make it more hydrophilicity prior to the MIPs modification [37]. However, for GCE or gold electrodes, it will be impossible because such operations can cause great damage to the electrodes. As most of the fundamental studies are completed via the GCE or gold electrode, we only introduce the MIECs based on the GCE or gold electrode operation system, so that the results can be compared under the same standard.

Regardless of the difference in electrodes, the MIECs can be divided into two groups according to different preparation methods of MIPs, which are (i) MIPs generated by an in-situ method such as electropolymerization of the monomer and (ii) the preformed MIPs will be coated on the electrode, also named as an ex-situ method [38,39]. For the in-situ method, MIPs can be directly obtained by electropolymerization without any processing requirement. The thickness of MIPs can be easily controlled by the applied current density and voltage. On the other hand, the ex-situ method for MIPs usually involves either spin-coating or spray-coating on the surface of the electrode as a thin film [40]. In the ex-situ approaches, preformed polymers on the surface of nanoparticles, especially the metallic nanoparticles are the widely used form, with the advantages of enhancing conductivity and mechanical stability.

### 2.3. The Construction of MIECs

MIECs are constructed by merging MIPs and electrochemical sensing platforms, and MIPs contribute to the function of specific recognition. Based on the electrode, MIPs are firstly synthesized through an ex-situ or in-situ polymerization method (Figure 3). Initiated by an inducer (catalyst, light, electricity, etc.), functional monomer produces polymers with molecule templates via covalent or non-covalent forces, and after polymerization of the crosslinker, the polymers are branded with the size and shape of template molecules [41,42]. After removing the template, the polymers remain at binding sites that are complementary to a template. By this means, a molecular memory is recorded by the polymer, in which the target molecule can rebind on the imprinted material with high specificity. Because of this, the MIPs can also be used in the sample pretreatment to separate and enrich the analytes.

Usually, MIPs behave with relatively low conductivity and mechanical stability due to the poor electron transport rate (ETR) and soft structure of polymers [43]. To enhance the specific surface area of the electrode and accelerate the ETR, some metallic nanomaterials such as gold nanoparticles (AuNP) are widely adopted in MECs [44]. Prior to the polymerization, solid metallic nanoparticles are firstly modified on the electrode to provide a new substrate for polymers, so that ex-situ or in-situ prepared MIPs could have a rigid skeleton and a flexible surface, which are favorable for both their mechanical stability and recognition flexibility. Meanwhile, for the design of signal output, additional signal molecules can be introduced in the MIPs as an internal reference signal. 

## 3. Applications of MIECs in Food Safety and Drug Detection

As a new kind of polymer material, MIPs have long been used in the field of analytical science. Compared with natural antibodies, MIPs can be easily modified and controllable in the affinity of receptor sites, which makes it more accessible to recognize different kinds of molecules, no matter small ones or big macromolecules [45,46]. In addition, MIPs show high stability, strong resistance to the environment and biocompatibility, which mean that they are of low cytotoxicity, and can be stored and used in harsh conditions, such as in low or high temperatures, extreme pH solutions, and strong ion strengths [47,48]. Because of their advantages, such as being lightweight, low-cost and easy to use, electrochemical sensors with a high diversity of electroanalytical techniques are expected to be the future generation of analytical systems. Taking the merits of MIPs and electrochemical sensors, MIECs behave promisingly in applications in food and drug safety detection (Figure 4).

### 3.1. Pathogen and Toxins

The ingestion of pathogen and toxin-contaminated food can cause severe illnesses, which pose a huge threat to human health [45,46]. A trace level of pathogen or toxin in the human body could inflict biological damage or even death [47]. Foodborne pathogens such as Salmonella enterica, Listeria monocytogenes, Escherichia coli, and Staphylococcus aureus, are responsible for poisoning food and water. Some mycotoxins produced by fungi, such as aflatoxins, fumonisins, ochratoxin A, and patulin can also induce physiological abnormalities in humans and animals, and most of them are tumorigenic [48,49]. Various analytical methods have been developed for the detection of pathogens and toxins in food samples, and MIECs have attracted much attention due to their high selectivity and sensitivity, as well as low cost and easy operation. 

For instance, Golabi et al. reported an electrochemical biosensor based on a whole-cell imprinting approach, which can deliver the rapid detection of *S. epidermidis* [50]. The cell-imprinted polymer with a boronic acid group endows a high affinity for bacteria, which was further used for the label-free detection of *S. epidermidis* via EIS with a linear response in the range of 10^3^–10^7^ CFU/mL. However, the presence of boronic acid groups will lead to non-specific absorption, making it less sensitive to the target. Thus, further studies can be carried out to eliminate the undesirable effects. Furthermore, Li et al. developed MIECs for Listeria monocytogenes (LM) based on 3-thiopheneacetic acid (TPA) as the functional monomer [51]. MIPs were prepared via the in-situ electropolymerization of TPA on the GCE surface in the presence of LM, which was denoted as LIP/GCE. In this case, [Fe(CN)_6_]^4−/3−^ is used as a probe to indicate the amount of LM. When LM cells are captured by LIP/GCE, the imprinted cavity will be filled up with LM and the access of [Fe(CN)_6_]^4−/3−^ to the electrode surface is blocked. As a result, the peak current of [Fe(CN)_6_]^4−/3−^ decreases with the increasing LM concentration. Finally, the MIECs behaved at a low limit of detection (6 CFU/mL) and a wide linear range (10 to 10^6^ CFU/mL). 

In addition, Guo et al. constructed MIECs for the determination of patulin based on electropolymerization technology with modifications of carbon dots, chitosan, and Au NP [52]. Wherein, 2–oxindole was adopted as a template to replace patulin and form the molecularly imprinted cavity at a lower cost, and the modifiers were used to increase electroactive areas and acquire distinct signals. The MIECs showed a linear range from 1 pM to 1 nM with the limit of detection (LOD) of 0.757 pM. Munawar et al. fabricated an ex- situ MIPs for the electrochemical detection of fumonisin B_1_ (FB_1_) [53]. The MIPs were firstly prepared and then covalently attached to the working electrode. With the probing redox couple of [Fe(CN)_6_]^4−/3−^, this sensor was allowed to detect FB_1_ via the impedimetric or voltammetric technique. The EIS and DPV techniques behaved in a linear detection range from 1 fM to 10 pM with LODs of 0.03 and 0.7 fM, respectively.

With the aid of MIPs, MIECs can easily achieve specific and sensitive detection of pathogens and toxins in food samples. However, the cell imprinting strategy still suffers from partial non-specific recognition, which requires further improvement in the blotting templates and preparation methods. On the other hand, though the electropolymerization strategy is convenient to prepare MIPs, functionalized nanomaterials are still needed to enhance the conductive properties of the electrodes.

### 3.2. Pesticide Residue

Pesticides are favored in agriculture for crops and seed protection, as they help raise the output of agricultural products. Though the use of pesticides can produce significant market prospects and huge social benefits, the pesticide residues in food materials also have deleterious effects on human health [54,55]. Most of the used pesticides and their residues have long-term stability and biological effects due to their high persistence in the environment [56]. For months or years, the toxicity and potential carcinogenicity of the pesticide residues still exist [57,58]. So, the pesticide residues are easy to accumulate in the human body through the food chain. To ensure food safety for consumers, it is vital to develop sensitive and effective methods for pesticide residue detection. MIECs could be a potential candidate for pesticide residue monitoring.

As an instance, Dai et al. developed novel MIECs for selective and sensitive detection of imidacloprid residues [59]. With the strategy of dual-template molecularly imprinted polymers (DMIPs), two different templates and thionine (TH) were electropolymerized, and TH with redox peak acts as an internal signal. For the two templates, one is a non-electroactive template for a single signal, and another is an electroactive template for dual signals. Thus, non-electroactive bensulfuron-methyl (BSM) and electroactive imidacloprid (IMI) can be detected with different modes: the occupation of the imprinted cavities with BSM indicates a single-signal output of TH, and that of IMI shows an on-off ratiometric signal. By this means, the sensor behaved in wide linear ranges of 10 nM–10 μM and 0.1 Μm–0.1 mM, with LODs of 7.8 × 10^−9^ M and 6.5 × 10^−8^ M for BSM and IMI, respectively. This study demonstrated the feasibility of DMIPs coupled with electrochemical techniques in the analysis of pesticide residues, which provided a new idea to construct selective MIECs for electroactive and non-electroactive template detection.

Based on Co_3_O_4_ nanowire and core-shell Co_3_O_4_@MOF-74 nanocomposite, Karimi-Maleh et al. developed a MIECs method for fenamiphos (FEN) analysis [60]. After the modification of nanocomposites on pretreated carbon electrodes, the preparation of MIPs was carried out by the in-situ electropolymerization with pyrrole as monomer and FEN as a template. The MIECs behaved in a linear detection range from 0.01 nM to 1.0 nM with a limit of quantification (LOQ) and LOD of 1.0 × 10^−11^ M and 3.0 × 10^−12^ M, respectively. Co_3_O_4_@MOF-74 nanocomposites provided a high surface area and fast electron transfer rate in the detection. However, the synthesis of nanocomposites with a high temperature may increase the cost, so it would be better to develop greener and more energy-efficient nanomaterials.

As discussed above, MIECs can achieve specific and sensitive detection of pesticides. Still, there are some limitations with MIECs. For example, with the popularization of pesticides, different pesticides often exist in one food sample. It is difficult for MIECs to detect multiple pesticides at the same time. Therefore, it can be an alternative to exploring the MIPs with diverse functional monomers.

### 3.3. Heavy Metal Ions

Heavy metal ions are commonly found in wastewater and classified as water pollutants. However, even in the soil, heavy metal ions are dangerous because they can be adsorbed by crops, fruits, and vegetables [61]. For instance, heavy metals such as mercury, lead, cadmium, chromium and arsenic are non-degradable and ubiquitously distributed, and they are considered hazardous compounds even at a low concentration [62]. With the intake of those heavy metal ions, people could suffer from enzyme inhibition, oxidative stress and impaired antioxidant metabolism [63]. Hence, it is critical to develop suitable techniques for the fast and accurate detection of metal ions. Most heavy metal ions are electrically active and highly susceptible to exchange electrons and produce characteristic electrochemical signals. MIECs can be the appropriate tool for the detection and quantification of heavy metal ions.

To impart selectivity of electrochemical sensors, modifiers with a strong affinity are commonly used to recognize target ions. For example, Motlagh et al. prepared a novel nanostructured cadmium(II) ion-imprinted polymers (IIPs) by a sol-gel process [64]. For the preparation of Cd-IIP materials, an ex-situ surface imprinting strategy combined with a sol–gel process was adopted to fabricate the carbon paste electrode. The electrochemical sensor was employed for voltammetry detection of Cd(II) with a linear range of 0.5–40 μg L^−1^ and LOD of 0.15 μg L^−1^. Similarly, Sebastian et al. adopted Pb(II) ions as a template to prepare IIPs by modifying multiwalled carbon nanotubes (MWCNT) on Pt electrodes [65]. The sites imprinted in MWCNT/IIPs are highly selective to Pb(II) ions and the Pt electrode showed a sensitive response with the modified nanostructure. CV and DPV tests were conducted to discuss the features of the IIP electrochemical sensor. The sensing system behaved with an LOD of 2 × 10^−2^ μM for Pb(II) ions, revealing promising applications in the detection of environmental and food samples.

Similar to MIPs, the IIPs can impart electrochemical techniques with selectivity and simplicity in the detection of real samples. However, IIPs may suffer some limitations including low binding capacity, irregular shape, poor target site accessibility, and heterogeneous binding site distributions. To avoid the limitations indicated, new technologies should be developed to prepare IIPs with good accessibility, high affinity, and selectivity to the target ions. Due to their high porosity, adjustable structures, and good stability, metal–organic frameworks (MOFs) can be used as efficient substrates to prepare IIPs and construct electrochemical sensors [66].

### 3.4. Antibiotics Monitoring

Antibiotics are a class of antimicrobial compounds that are widely used in human or veterinary medicine to treat diseases, especially in the livestock industry and aquaculture [67]. However, the abuse of antibiotics could result in sustainable adverse effects on human health and the environment. The constant intake of antibiotics could cause immunopathological effects, hepatotoxicity, carcinogenicity, bone marrow toxicity, reproductive disorders, or even anaphylactic shock [68]. When antibiotics enter the water and land environment, the cycle of water will make it a significant local point of contamination. The overdosage of antibiotics in animals can lead to antibiotic residues in foodstuffs such as meat, chicken, egg, milk, and fish [69]. Through the enrichment of the food chain or the transfer of water, the antibiotics will finally accumulate in the human body and pose potential risks to human health. Therefore, it is imperative to develop effective methods of monitoring the antibiotic residues.

Paracetamol (PR), a kind of analgesic, antipyretic and anti-inflammatory drug, and is one of the most commonly consumed pharmaceuticals. In a normal dose, PR does not produce any harmful side effects, however, overdose intake of the drug could cause pancreas inflammation or even kidney damage and hepatotoxicity [70]. For instance, Dai et al. reported MIECs for the detection of PR based on Prussian blue (PB) embedded MIPs as a reference signal [71]. Herein, the inner layer of PB acts as an internal electrochemical signal and the target PR as another signal. When PR molecules were captured and incorporated with the cavity on the outer layer of MIPs, the redox current of PR increased while that of PB decreased due to the occupied sites’ blocked electron transfer, which finally manifested as an “on and off” signal output mode. As a result, the sensor displayed a concentration range from 1.0 nM to 0.1 mM with a LOD of 0.53 nM, as well as recoveries in the range between 94.6 and 104.9%, revealing it is acceptable in the practical applications.

Detection resolution has been identified as an important factor in newly developed analytical techniques, which reflect their ability to distinguish the details of analytes with similar structures. For instance, propranolol (prop), an important and widely used β-adrenaline antagonist for the treatment of cardiovascular diseases, has a similar chemical structure to salbutamol [72]. Moreover, prop has two enantiomers of S-prop and r-prop, and only S-prop has pharmacological performance. Based on the reduced graphene oxide (rGO) and chitosan-based MIPs, Liu et al. developed a differential potential ratio sensing platform for binary molecular recognition of prop [73]. In the platform, MIPs specifically recognize and capture prop enantiomers, rGO acts as a conductive substrate to produce an amplified signal, and the potential difference between the R-/S-prop offers the ratiometric signal. As a result, the method gained a distinct potential difference of 135 mV with a detection range from 50 μM to 1000 μM in the racemic mixture, which reveals great potential in the fields of pharmacological detection and clinical analysis.

Chlorpromazine (CPZ) is an antipsychotic drug used to treat psychiatric and personality disorders, and clinical monitoring of CPZ is necessary. Liu et al. presented Pt/Co_3_O_4_ nanoparticles and methylene blue (MB) monomer-based MIPs for selective detection of CPZ [74]. Wherein, MB molecule in MIPs provides a fixed internal signal, and the signal of CPZ changes with concentrations, which is a typical on/off ratiometric signal output mode. Under optimal conditions, the method showed a linear range of 0.005–9 μM with a LOD of 2.6 nM and recoveries of 95.3–108.0% in pharmaceutical samples. The dual-signal output mode provides built-in signal calibration to eliminate interference and adjusts the signal fluctuation, thus can effectively improve detection stability and accuracy.

Besides food safety and drug detection, MIECs combined with nanomaterials have also shown good performance for the screening of biomarkers. As biomarkers are closely related to some diseases, it is important to achieve stable and sensitive detection of biomarkers for the early diagnosis. To fulfill the rapid screening, comprehensive procedures must be taken to treat the complex media, which usually involves preconcentration and separation. MIECs can take the role to realize the separation and detection at the same time. For example, Anirudhan et al. reported MIECs for the detection of 2-aminoadipic acid (2-AAA), a diabetes biomarker, based on the surface modification of electrode with a drop-casting method [75]. The modified MIP electrode showed good DPV results for 2-AAA with an LOD of 0.40 × 10^−11^ M, demonstrating the high selectivity and sensitivity of the MIECs for real sample analysis.

## 4. Discussion

MIECs are a class of newly developed sensing methods that combine the sensitivity of electrochemical techniques and the selectivity of MIPs. The most important part of constructing MIECs is to prepare MIPs-based electrodes, which can be regarded as a form of electrode modification with MIPs, which include the in-situ electropolymerization and ex-situ polymerization and modification. For the electropolymerization method, it can directly synthesize MIPs with electroactive monomers on the electrode by applying an appropriate potential or performing CV scans. On the other hand, the ex-situ modification of electrodes can be conducted by incubating preformed MIPs on the electrode surface, which can be more flexible to meet the requirements for different applications. However, the in-situ method can obtain an entire molecularly imprinted electrode without worrying about the shed of MIPs from the electrode, which means the in-situ MIPs could exhibit a more accurate and stable performance. In the detection process, redox products will be generated and get fouled on the electrode surface, which could block the reversibility of binding on the transducer surface. MIPs can decrease the foul of the electrode due to the analyte species being freely diffused in the pores or channels of MIPs.

According to the electrochemical properties of MIPs, different signals (CV, EIS, DPV, SWV, etc.) can be chosen to detect the analytes. To build different signal output modes, the detection strategies can be divided into a single signal and dual singles, and the dual-signal output mode includes on/off and on-ff modes, which can provide a built-in correction factor to eliminate the interferences and improve stability. As the single signal readily changes with the external conditions, dual ratiometric signals are widely accepted as an effective strategy to improve the detection stability and accuracy. In fact, electrochemical detection is a process of surface reaction with the participation of electrons. In the presence of MIPs layers, analytes can be specifically recognized through the pores or channels and then detected with oxidized peaks. MIPs surface also acts as a film for recognition and reaction, which can be served as a platform for studying the mechanisms such as reaction kinetics.

This review mainly presented the MIECs from the preparation of MIPs to various applications. To better understand the performance of MIECs in different analytical fields, Table 1 summarized detection parameters such as relative standard derivation (RSD), linear detection range and LOD, revealing their good stability and sensitivity. 

## 5. Conclusions

This review specified recent advances and applications of several newly developed MIECs in food and drug safety. A growing number of research papers related to MIECs demonstrate that MIPs coupled with electrochemical sensors outcompete traditional electrochemical sensors in selectivity and stability. MIPs have been successfully prepared on electrodes to specifically capture analytes with electrochemical signals. Meanwhile, the MIP techniques can achieve one step of separation and detection with simple procedures and easy operations. In addition to MIPs preparation, the signal output can be designed to obtain enhanced detection robustness. For example, the dual-signal output mode is widely used as an effective strategy to avoid signal fluctuation, and thus can improve detection performance such as anti-jamming ability and reproducibility. In this regard, MIECs open a new avenue to enhance the detection performance of traditional electrochemical methods. 

Despite significant breakthroughs in the design and construction of MIECs, it still remains a great challenge to make MIECs a powerful tool to meet the requirements in practical applications. For instance, the molecularly imprinted electrodes are still prone to fouling, and their surfaces are difficult to strip for further use. Furthermore, the detection mode is still dependent on the traditional electrode system, which makes it difficult to achieve effective and high-throughput detection. At present, there is no widely accepted standard in practice for constructing electrochemical sensors, so the detection reproducibility could not be guaranteed. To overcome these limitations, the preparation of MIPs must be improved to obtain a more homogeneously binding site population with a high affinity for the target analyte. It can be another way to break the limitations by merging MIECs with nanotechnology to construct new nanobiosensors [76]. It is urgent to improve the protocols of MIECs with specificity, selectivity, and sensitivity in commercial applications. As the electrochemically synthesized MIPs are complex and versatile, it is also important to study the mechanism and set common criteria for the preparations. Considerable effort has been devoted to dealing with the above problems. We believe that in the near future, the limitations of MIECs will be successfully addressed and the MIECs will occupy an important position in the sensor market.

## Figures and Tables

**Figure 1 biosensors-12-00369-f001:**
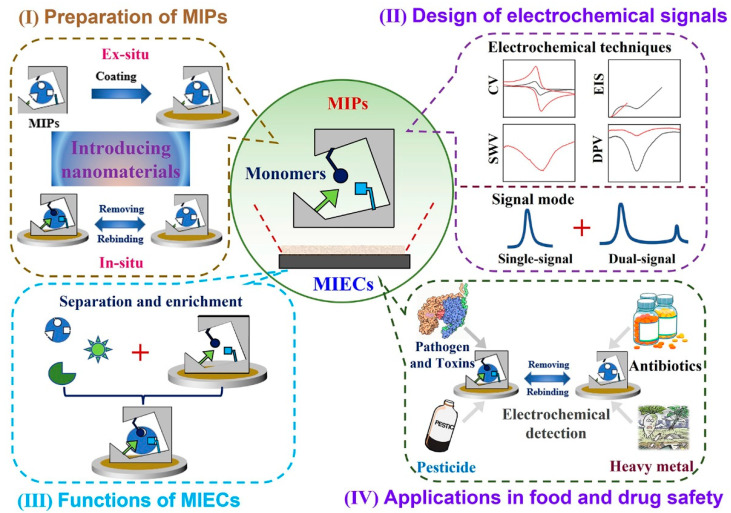
Schematic presentation of MIECs: (I) preparation of MIPs, (II) design of electrochemical signal, (III) functions of MIECs and (IV) applications in food and drug safety detection.

**Figure 2 biosensors-12-00369-f002:**
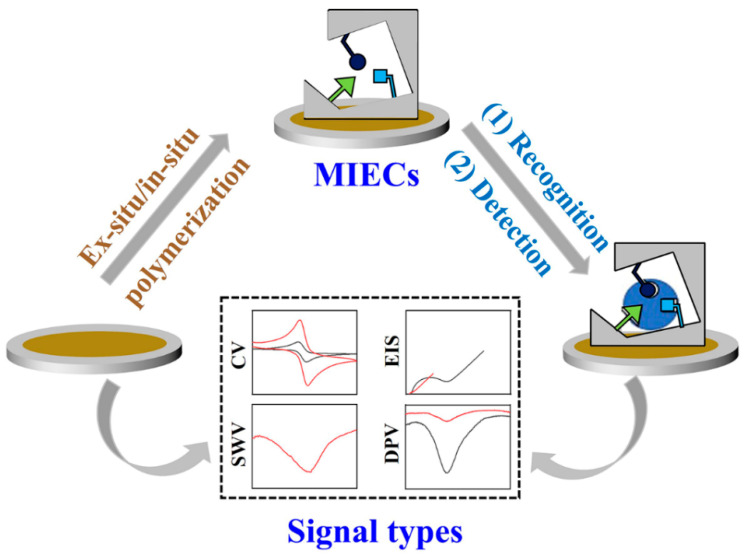
Illustration of working principle of MIECs from the preparation of MIPs to electrochemical signal output.

**Figure 3 biosensors-12-00369-f003:**
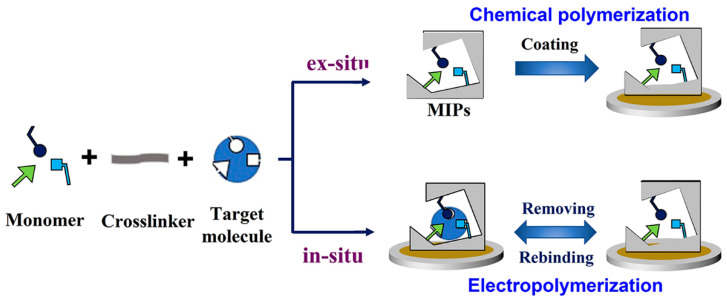
Schematic view of the construction of MECs through an ex-situ/in-situ method.

**Figure 4 biosensors-12-00369-f004:**
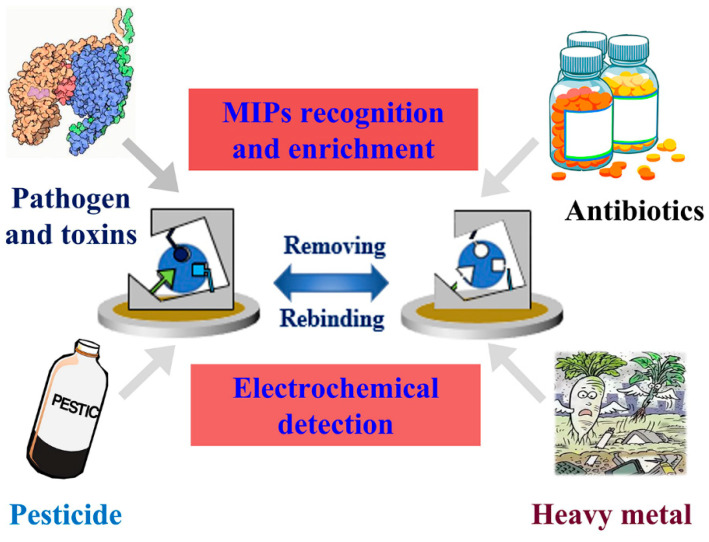
Schematic presentation of MIECs in the applications of food and drug safety detection.

**Table 1 biosensors-12-00369-t001:** Summary of MIECs methods in different applications (not given: N).

Type of Application	Test Target	R Value	RSD	Linear Range	LOD	References
Pathogen and toxins	*S. epidermidis*	0.9730	N	10^3^–10^7^ CFU/mL	7.5 × 10^−8^ M	[50]
	LM	N	N	10–10^6^ CFU/mL	6 CFU/mL	[51]
	Patulin	0.9953	7.3%	1 × 10^-12^–1 × 10^−9^ M	7.57 × 10^−13^ M	[52]
	FB1	0.9899	N	1 fM–10 pM	0.03 fM	[53]
		0.9798	N		0.7 fM	
Pesticide residue	IMI	0.9987	4.5%	1.0 × 10^−7^–1.0 × 10^−4^ M	6.5 × 10^−8^ M	[59]
	FEN	0.9995	N	1.0 × 10^−11^–1.0 × 10^−9^ M	3.0 × 10^−12^ M	[60]
Heavy metal ions	Cd(II)	0.9989	2.7%	0.5–40 µg L^−1^	0.15 μg L^−1^	[64]
	Pb(II)	0.9993	N	1–5 ppm	2 × 10^−2^ μM	[65]
Antibiotics monitoring	PR	N	1.2%	1.0 nM–0.1 mM	0.53 nM	[71]
	s/r-Prop	N	N	50 μM–1000 μM	N	[73]
	CPZ	0.9981	0.94%	0.005–9 μM	2.6 nM	[74]

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
