# Peer review of "Towards Development of Molecularly Imprinted Electrochemical Sensors for Food and Drug Safety: Progress and Trends"

_biosensors, 2022, doi:10.3390/bios12060369_

Round 1

Reviewer 1 Report

Journal: Biosensors

Article title: Towards development of molecularly imprinted electrochemical sensors for food and drug safety: Progress and trends

The authors have made review research “authors studied the applications of molecularly imprinted electrochemical sensors for food and drug safety”; it is an interesting review research article title and scope match the journal.

I recommend its publication with major revision.

  1. The abstract is very poor authors need to mention the drawbacks of MI electrochemical sensors how research had been carried out to overcome this
  2. Introduction needs to be further improved with a proper literature survey
  3. Resolution of Fig 1 has to be improved
  4. This review is centered around electrochemical sensing however there is hardly any detailed discussion or relevant images pertaining to the title. So the authors need to include figures
  5. corresponding to electrochemical characterization and bringing in discussion in the manuscript
  6. Please check the grammatical and syntax error
  7. There has to be a graphical abstract that can create curiosity for the readers and present the work aesthetically.

Author Response

The authors have made review research “authors studied the applications of molecularly imprinted electrochemical sensors for food and drug safety”; it is an interesting review research article title and scope match the journal.

Response: Thank you for the positive comments and constructive suggestions. We have revised the manuscript as suggested and the answer/description of the actions are listed as follows.

1. The abstract is very poor authors need to mention the drawbacks of MI electrochemical sensors how research had been carried out to overcome this

Response: Thank you for the suggestions. We modified the abstract and the limitations of MIECs was mentioned as suggested.

2. Introduction needs to be further improved with a proper literature survey

Response: Thank you for the suggestions. We have modified the Introduction as suggested. Actually, we aimed to explore different MIECs regarding to the design, working principle and functions. In the Introduction section, we introduced the reasons why we choose the MIECs in food and drug safety, as well as their challenges and prospects for developing new electrochemical methods.

3. Resolution of Fig 1 has to be improved

Response: Thank you for the suggestion. Fig 1 was modified as suggested.

4. This review is centered around electrochemical sensing however there is hardly any detailed discussion or relevant images pertaining to the title. So the authors need to include figures corresponding to electrochemical characterization and bringing in discussion in the manuscript

Response: Thank you for the suggestions. We have included scheme and figures in the text. Since this is a review article, we don’t have experimental data ourselves to discuss with. However, we do give critical comments and discussed the reported work as suggested.

5. Please check the grammatical and syntax error

Response: Thank you for the kind reminding. We carefully checked and revised the manuscript as suggested.

6. There has to be a graphical abstract that can create curiosity for the readers and present the work aesthetically.

Response: Thank you for the good suggestions. We have added a graphical abstract as suggested.

Reviewer 2 Report

Ref: biosensors-1726301

Title of the manuscript:Towards development of molecularly imprinted electrochemical sensors for food and drug safety: Progress and trends.”

In this review article, the authors have explained the use of molecularly imprinted electrochemical platforms for food and drug safety. The authors have explained various aspects of the principle, classification, and construction of MIP’s and their respective applications in various fields. The paper is quite enriched with information but the continuity between the paragraphs is missing, making it boring and voluminous to the readers. The authors should make it crisp by removing extra words and repetitions. The manuscript needs extensive English grammatical editing. In particular, many places require more details about the quality of testing using MIEC’s compared to both traditional and electrochemical approaches. A few points are highlighted below:

  1. There is attention required for English grammatical errors. More scientific terminologies should be added.

Line 37, “pose greatly effects”.

Line 41, “a great of work”.

Line 53, “thus will affecting”

Line 61. Line 84, “which make MIEC become more sensitive”.

Line 152, “which behave the advantages”.

And many others.

  1. The authors have not explained the setbacks of using MIEC-based electrochemical platforms. It’s fouling and other effects. Only one statement about the challenge for design and construction has been mentioned in Line 404. More information on its setbacks should be added.

  1. The introduction should include a section for the literature accounting for the use of conventional techniques for food and drug safety. And the rationale behind using the electrochemical platform instead of traditional approaches.

  1. There is a lacuna in the continuity of the manuscript.

Author Response

In this review article, the authors have explained the use of molecularly imprinted electrochemical platforms for food and drug safety. The authors have explained various aspects of the principle, classification, and construction of MIP’s and their respective applications in various fields. The paper is quite enriched with information but the continuity between the paragraphs is missing, making it boring and voluminous to the readers. The authors should make it crisp by removing extra words and repetitions. The manuscript needs extensive English grammatical editing. In particular, many places require more details about the quality of testing using MIEC’s compared to both traditional and electrochemical approaches. A few points are highlighted below:

Response: Thank you for the comments and constructive suggestions. The manuscript has been carefully revised as suggested and the modifications are highlighted in red in the revised version of the manuscript.

1. There is attention required for English grammatical errors. More scientific terminologies should be added.

Line 37, “pose greatly effects”.

Line 41, “a great of work”.

Line 53, “thus will affecting”

Line 61. Line 84, “which make MIEC become more sensitive”.

Line 152, “which behave the advantages”.

And many others.

Response: Thank you for the kind reminding and suggestions. We carefully checked the manuscript, and the errors have been revised as suggested.

2. The authors have not explained the setbacks of using MIEC-based electrochemical platforms. It’s fouling and other effects. Only one statement about the challenge for design and construction has been mentioned in Line 404. More information on its setbacks should be added.

Response: Thank you for the suggestions. More information on its setbacks has been added in the test as suggested (line 419-424).

3. The introduction should include a section for the literature accounting for the use of conventional techniques for food and drug safety. And the rationale behind using the electrochemical platform instead of traditional approaches.

Response: Thank you for the suggestions. We added some sentences for the use of conventional techniques for food and drug safety and revised the manuscript as suggested.

4. There is a lacuna in the continuity of the manuscript.

Response: Thank you for the kind reminding. We have carefully revised the whole manuscript and some parts are rewritten to make it more smooth for readers.

Reviewer 3 Report

This review summarizes all kinds of MIECs as well as the preparation of MIPs, design of electrochemical signals, functions of MIECs and applications in food and drug safety, aiming to present a general comment on the development of MIECs and make a bridge between MIPs and electrochemical sensors. The paper well writing with interesting schemes but I am a little concern with novelty of paper and little discussion. Authors need to address following issues:

-there are many similar reviews in this issue, what is the novelty of your review?

-please add sensor to the keywords

-the review should be discussed following refs for better comparison: https://doi.org/10.1016/j.tifs.2021.10.024, https://doi.org/10.1016/j.foodchem.2020.127797, https://pubmed.ncbi.nlm.nih.gov/35173432/, https://doi.org/10.1016/j.ijbiomac.2022.02.082

- some abbreviation used without explanation

-some refs are out of dated and need to be replaced with new ones

Author Response

This review summarizes all kinds of MIECs as well as the preparation of MIPs, design of electrochemical signals, functions of MIECs and applications in food and drug safety, aiming to present a general comment on the development of MIECs and make a bridge between MIPs and electrochemical sensors. The paper well writing with interesting schemes but I am a little concern with novelty of paper and little discussion. Authors need to address following issues:

1. there are many similar reviews in this issue, what is the novelty of your review?

Response: Thank you for the question. As a matter of fact, few reviews have been published on the topic of MIECs from the perspective of integrated MIPs and signal output. In this review, we highlighted different types of MIECs as well as their design, working principle and applications. Additionally, we discuss the current challenges and prospects of MIECs to provide innovative idea tactics for developing MIECs, which could help the reader clearly know what the MIECs can do, how to do it and where it can go.

2. please add sensor to the keywords

Response: Thank you for the suggestions. We have added sensors to the keywords as suggested.

3. the review should be discussed following refs for better comparison: https://doi.org/10.1016/j.tifs.2021.10.024, https://doi.org/10.1016/j.foodchem.2020.127797, https://pubmed.ncbi.nlm.nih.gov/35173432/, https://doi.org/10.1016/j.ijbiomac.2022.02.082

Response: Thank you for the suggestions. We have discussed the above references as suggested (line 31-33, line 426-427).

4. some abbreviation used without explanation

Response: Thank you for the reminding. The abbreviations have been explained suggested.

5. some refs are out of dated and need to be replaced with new ones

Response: Thank you for the suggestions. Some of the references are replaced as suggested.

Round 2

Reviewer 1 Report

The authors made sufficient changes and the article can be accepted in the present format 

Author Response

The authors made sufficient changes and the article can be accepted in the present format.

Response: We gratefully acknowledge the Reviewers’ comments that highly contributed to the improvement of the considered manuscript.

Reviewer 2 Report

Ref: biosensors-1726301

Title of the manuscript: Towards development of molecularly imprinted electrochemical sensors for food and drug safety: Progress and trends.”

Authors have worked on the revision of the manuscript but still lack the essential English editing. Authors should incorporate proper scientific terms instead of just basic English terminologies. A few important additions are required:

1.      The authors should check the literature for the comparison of traditional and electrochemical approaches for food and drug safety. The traditional techniques for food and drug safety accounts using optical sensors, electronic noses, aptamer-based, etc. A brief comparative analysis is required to be included in the introduction.

2.      Authors should confine a table accounting for various sensing platforms (traditional and electrochemical) vs. MIEC’s based techniques for their merits and demerits.

3.      The role of bonding, reactivity, and complementary binding sites should be explained. The chemistry behind the sensors should be included.

4. English editing required Line 34, 35, and 38. Spelling mistake Line 48 (caried). Use scientific terminologies example: Line 57 “it will produce a low reproducibility.” 

Author Response

Authors have worked on the revision of the manuscript but still lack the essential English editing. Authors should incorporate proper scientific terms instead of just basic English terminologies. A few important additions are required:

1. The authors should check the literature for the comparison of traditional and electrochemical approaches for food and drug safety. The traditional techniques for food and drug safety accounts using optical sensors, electronic noses, aptamer-based, etc. A brief comparative analysis is required to be included in the introduction.

Response: Thank you for the suggestions. For the traditional electrochemical approaches, we have discussed many literatures in paragraphs 2 and 3 to explain the existed problems in electrochemical sensors from four aspects. For optical sensors, electronic noses, aptamer-based techniques, they are not the topic in this work, and these methods are different from electrochemical sensors in working principle, construction structure, signal output, or even the transduces, so we believe that it is not appropriate to add these methods in the Introduction part. Thank you for the kind understanding.

2. Authors should confine a table accounting for various sensing platforms (traditional and electrochemical) vs. MIEC’s based techniques for their merits and demerits.

Response: Thank you for the suggestion. Here I would like to give some explanations. As a matter of fact, as mentioned by the other two referees, the drawbacks of electrochemical sensors and the possible solutions were added and discussed in the General summary. Besides, this is a review article but not a research article, our aim is to discuss the current challenges and prospects of MIECs to provide innovative idea tactics for developing electrochemical sensors, which could help the reader clearly know what the MIECs can do, how to do it and where it can go, rather than demonstrating the superior performance of MIECs than other methods. So, we didn’t list the table and compared them for their merits and demerits. Thank you for the kind understanding.

3. The role of bonding, reactivity, and complementary binding sites should be explained. The chemistry behind the sensors should be included.

Response: Thank you for the suggestion. I don’t mean to offend you that I was confused by the question. I really don’t understand about what the bonding, reactivity, and complementary binding sites mean. For the construction and structure of MIECs, we have discussed in the manuscript, but we don’t mention anything about bonding and reactivity in the manuscript.

4. English editing required Line 34, 35, and 38. Spelling mistake Line 48 (caried). Use scientific terminologies example: Line 57 “it will produce a low reproducibility.” 

Response: Thank you for the kind reminding. We checked the whole manuscript, and the English edits were performed as suggested.

Reviewer 3 Report

The paper can be accepted

Author Response

The paper can be accepted.

Response: Thank you very much for the Reviewers’ comments that highly contributed to the improvement of the considered manuscript.

Round 3

Reviewer 2 Report

After required English editing, the manuscript can be recommended for publication.